# Bibliometric Analysis and Literature Review of Tourism Destination Resilience Research

**DOI:** 10.3390/ijerph19095562

**Published:** 2022-05-03

**Authors:** Tian Wang, Zhaoping Yang, Xiaodong Chen, Fang Han

**Affiliations:** 1State Key Laboratory of Desert and Oasis Ecology, Xinjiang Institute of Ecology and Geography, Chinese Academy of Sciences, Urumqi 830011, China; wangtian181@mails.ucas.ac.cn (T.W.); chenxiaodong181@mails.ucas.ac.cn (X.C.); hanfang@ms.xjb.ac.cn (F.H.); 2University of Chinese Academy of Sciences, Beijing 100049, China

**Keywords:** tourism destination resilience, CiteSpace, bibliometric analysis, research hotspots, future research directions

## Abstract

The application of resilience thinking to tourism destination research is a new perspective on sustainable tourism and has gradually become a popular research topic. Some literature has been conducted on tourism destination resilience, but there has not been a comprehensive review and analysis of the whole field. This study was based on the literature from 2000 to 2021 in the Web of Science core collection database. The collaboration analysis, literature co-citation analysis, keyword co-occurrence, burst detection analysis in CiteSpace, and qualitative analysis were adopted to conduct a holistic tourism destination resilience research review. The results indicated that the United States, Australia, China, and the United Kingdom were the primary countries involved in tourism destination resilience research. Five hot research themes were obtained. (1) concept and connotation of tourism destination resilience, (2) drivers of tourism destination resilience, (3) sustainable management framework and practices, (4) perception of tourism destination resilience, and (5) the resilience of the tourism community. Furthermore, four research gaps and future directions were proposed in this study, including the theoretical framework of tourism destination resilience, assessment of tourism destination resilience, sustainable management and resilience, and application of advanced technology in tourism destination resilience. This study assists researchers in understanding the development and future research directions in tourism destination resilience research.

## 1. Introduction

With increased uncertainty in the natural and social environment and the increased frequency of disasters and crises, building resilience has become an effective way to promote sustainable development [1]. Resilience was initially defined as “the ability of a system to absorb disturbances and reorganize in response to changes to maintain substantially the same functions, structures, attributes, and feedback as before the disturbances occurred [2]”. Subsequently, the resilience of the social-ecological system was emphasized to reach a new state. Resilience is not a new concept, but its introduction into tourism research is still nascent [3].

In recent years, economic crises, terrorist attacks, earthquakes, and other social crises and natural disasters have threatened tourism destinations, especially the COVID-19 epidemic since 2020. In 2020, 100% of destinations worldwide had implemented travel restrictions, and 27% had closed their borders entirely to international tourism [4]. As an essential basis for crisis management and sustainable development in tourism destinations, destination resilience research has become a focus of intergovernmental organizations and academia [5,6].

Tourism destination resilience is the ability of tourism destinations to resist, adapt and self-organize against disturbances. It is a new way and perspective for tourism destinations to cope with threats posed by various natural or human-induced crises and uncertainties [7,8]. Although there is no uniform definition of tourism destination resilience, researchers have reached a consensus to shift the perspective from crisis management to resilience research in tourism studies [9]. Enhancing the resilience of tourism destinations can be helpful for tourism destinations to better adapt to changes and achieve sustainable development. Hence, it is urgent and vital to strengthen the research on tourism destination resilience.

With the introduction of resilience into tourism research and the growing prominence of sustainable development issues in tourism destinations, research on tourism destination resilience has been enhanced. Since the 21st century, scholars have focused on tourism destination resilience research, and research outputs have emerged especially over the last decade. There have been some research advances on tourism destination resilience in concepts, frameworks, management, and assessment. Several reviews were conducted on crisis management in tourism destinations. To some extent, tourism destination crisis management is also part of destination resilience. For example, Ritchie [10] reviewed papers published in tourism risk, crisis, and disaster management from 1960 to 2018 and critically analyzed three research themes. Jiang [11] used CiteSpace to analyze and visualize the knowledge structure in tourism crisis and disaster management and found that tourism crisis and disaster management research has shifted from broader themes to more specific issues, most recently focusing on resilience. However, although previous reviews have reviewed tourism destination resilience research from some specific perspectives, a systematic and comprehensive analysis of this research area seems to be lacking.

It is necessary and meaningful to conduct a holistic and systematic review of past research, especially at a time when destination resilience research is still emerging. Therefore, this study aims to systematically summarize the research content, generalize the main research strengths and research hotspots, the gaps, and the future research directions on tourism destination resilience. Specifically, this study hopes to solve the following three research questions in the research of tourism destination resilience. RQ1. What are the research strengths and their collaborations on tourism destination resilience during the past two decades? RQ2. What are the hot research themes in tourism destination resilience? RQ3. What are the research gaps and future research directions in tourism destination resilience?

The paper is structured as follows. The first part is an introduction to the resilience of tourism destinations and existing research. Next, the Section 2 describes the analysis methodology and data collection. Then, the Section 3 analyzes the bibliometric results, including collaboration network analysis, literature co-citation analysis, keyword co-occurrence analysis, and burst detection analysis. The Section 4 discusses the analysis results in depth and describes hot research themes, research gaps, and future research directions. Finally, the Section 5 summarizes the findings and implications of this study and discusses shortcomings and prospects for future research.

## 2. Materials and Methods

### 2.1. Scientific Knowledge Mapping

Scientific knowledge mapping, also known as knowledge domain visualization, visualizes the development process and structural relationship of knowledge with the knowledge domain as the object [12,13]. It is intuitive and efficient to adopt knowledge mapping to analyze the research hotspots and frontiers in a specific field. Generally, there are four steps in scientific knowledge mapping [14]. The first step is to build a base database by downloading relevant documents on specific topics from literature databases. Then, the second step is to utilize the bibliometric software to develop the co-occurrence, co-citation, and co-authorship matrices. The third step is data visualization. Finally, the fourth step is to conduct an in-depth analysis of the scientific knowledge map with background knowledge. Some software has been available for building knowledge mapping. Moreover, the knowledge mapping in this study was implemented on CiteSpace 5.8 R3, for reasons described in Section 2.2.

### 2.2. Data Analysis Method

CiteSpace is a knowledge mapping software based on quantitative analysis developed by Prof. Chaomei Chen [12]. CiteSpace has been widely adopted by researchers from various fields for its powerful features and effective visualization [15,16,17]. This study mainly used collaboration network analysis, co-citation analysis, keyword co-occurrence analysis, and burst detection analysis in CiteSpace [14].

The collaboration networks contribute to the analysis of the distribution of research strength and their collaboration in a particular research field. This analysis aims to facilitate potential research collaborations better. CiteSpace provides scientific network analysis at the macro, medium, and micro levels, including national/regional, institutional, and author collaboration networks [14].

The concept of co-citation was first introduced by Henry Small, an American intelligence scientist, in 1973 [18]. Co-citation analysis is a highlight feature that distinguishes CiteSpace from other metrics software, which helps researchers track the evolution of the research field [19]. Clustering analysis of literature co-citation allows exploration of common themes in similar literature.

The co-word analysis is based on the frequency distribution of word occurrences. The co-occurrence analysis tools in CiteSpace include keyword co-occurrence, term co-occurrence, and category co-occurrence. In this study, keyword co-occurrence analysis and burst detection analysis was used [14]. Keywords are a highly concise summary of the content of the literature. Keyword co-occurrence and burst detection analysis are essential tools for identifying research hotspots and development trends in the research field.

### 2.3. Material

Data collection is the basis of literature analysis. Publications in academic journals can reflect the cutting-edge dynamics of academic research, so it is essential to obtain abundant literature for the literature review. The Web of Science (WoS) core collection databases (Social Science Citation Index (SSCI), Science Citation Index Expanded (SCIE)) were selected as data sources for this study. WoS database is a worldwide recognized comprehensive database containing authoritative and influential journals. The authority of the WoS database in bibliometrics has been proven by many studies, including those in the field of tourism research [20,21]. Hence, the selection of the WoS database for this study will guarantee the reliability of the data.

Several search criteria were required to be set in WoS to screen the literature related to the study topic. This study identified the topic as destination resilience, with the search formula: TS = (touris* AND resilience) or TS = (destination AND resilience). The time span was set to “2000–2021”. Because research on destination resilience was scarce and not very relevant before the 21st century, it began to develop gradually after the 2000s. The language was set to English. Only journal articles and reviews were collected to search for more influential literature. The search was conducted on 27 February 2022, and the database was last updated on 27 February 2022, with a total of 310 documents collected. The search results included some literature that was not relevant to tourism destination resilience, so it was necessary to conduct a second round of screening to exclude the irrelevant results manually. Finally, 207 valid results were retained as the research sample.

## 3. Results

### 3.1. Overview

The annual number of publications serves as a good indicator of evolutionary tendency in the research field. The analysis of the number of publications per year is also the basis of the bibliometric analysis [22,23]. There was a continuous growth trend in the number of publications on tourism destination resilience from 2000 to 2021, indicating an increasing interest of scholars in tourism destination resilience research (Figure 1). Until 2012, the average number of publications was below ten annually, with a slight and flat interannual variation. From 2012 to 2018, the number of publications showed a fluctuating upward trend.

In contrast, the number of publications increased significantly and rapidly in 2019–2021. It is worth mentioning that the number of publications has reached 68 in 2021, which is four times more than in 2018. Such a positive trend of continued growth dramatically indicated that destination resilience research would remain promising in the future.

### 3.2. Collaboration Analysis

#### 3.2.1. Countries and Regions

There were 66 nodes and 127 links in the collaboration network between countries or regions (Figure 2). Regarding the continental distribution of countries or regions, Europe, Asia, Oceania, and North America have been relatively well studied for destination resilience. On the other hand, Africa and the North and South continents were relatively weak. Judging from the ranking of the number of publications by country or region, the United States (44), Australia (35), China (28), and the United Kingdom (24) have relatively more research on this topic, with the United States accounting for 13.37% of the publications. Regarding centrality, the United Kingdom, the United States, Australia, and New Zealand ranked in the top four, indicating that these four countries were more cooperative externally than others, shown by the purple ring in Figure 2. Remarkably, although China has published much research on tourism destination resilience, the low centrality demonstrated that there is still a lack in the degree of international cooperation in China. This is essential to strengthening external collaboration and communication in the future. By the end of 2021, 66 countries or regions had participated in tourism destination resilience research, accounting for roughly a third of the total number of countries and regions worldwide, revealing the lack of attention given to tourism destination resilience research in the majority of countries and regions.

#### 3.2.2. Institutions and Authors

From the perspective of research institutions, 220 research institutions have conducted tourism destination resilience research, of which they have collaborated 241 times. The network density of 0.01 suggested, to some extent, that there is significant potential for collaboration between research institutions. It was interesting to note that the top three institutions, the University of Queensland, James Cook University, and Massey University, were all located in the New Zealand region. In contrast, the other research institutions were scattered around the world. Furthermore, the University of Queensland and James Cook University’s centrality were higher than the other institutions, demonstrating that these two universities demonstrate significant research power in collaborative research on destination resilience.

The density of the author collaboration network was 0.0082, reflecting the relative lack of collaboration among researchers. There were 269 researchers in the field of tourism destination resilience. The studies of BRENT W RITCHIE, CAROLINE ORCHISTON, and C MICHAEL HALL have been cited relatively more frequently, suggesting that the studies of these scholars have received relatively high recognition and attention. The academic backgrounds of these scholars are complex, covering various fields such as tourism, economics, management, and disasters. Thus, it reflects, to some extent, the multidisciplinary characteristics of tourism destination resilience research.

### 3.3. Co-Citation Analysis

Literature co-citation analysis is a crucial method for tracking the frontiers of science and the research base, and its effectiveness has been demonstrated in a variety of studies [24,25]. In this study, the co-citation of tourism destination resilience research was clustered and analyzed with the more commonly employed LLR algorithm [21]. The nodes in the co-citation network represented the cited literature, and the links between the nodes described the co-citation relationships between the literature. There were 567 nodes and 1550 links in the co-citation network, and 12 clusters were generated. The modularity Q value of the network was 0.8832, which is greater than 0.3, implying that the clustering structure is significant (Figure 3). The silhouette values ranged from 0.838 to 1, and the weighted mean silhouette value was 0.9195, demonstrating that the clustering results were convincing.

Cluster #0 “destination recovery” is the largest cluster containing 67 members. The tourism industry has contributed to economic and sustainable development, and tourism destinations are a vital component of tourism. However, tourism destinations are significantly vulnerable to disturbances. Thus, destination recovery has become an essential issue in tourism destination research. After suffering the disturbances, do tourist destinations return to their original state, or do they reach a new equilibrium? This question has attracted attention [1,26]. Resilience plays a vital role in the recovery of tourism destinations and is the goal of destination recovery [27]. Therefore, the discussion of the goals of destination recovery provides an insight into the connotation of tourism destination resilience. Most studies have concluded that destination recovery prefers destinations to reach a healthier state of systemic equilibrium after recovery, revitalizing them. The representative literature of this research cluster constructed a multidimensional framework of destination recovery based on vulnerability, resilience, and adaptation, and analyzed the factors affecting resilience and adaptation [1].

Cluster #1 “regional tourism destinations” and Cluster #2 “tourism destinations” are related to tourism destinations. Researchers have studied the resilience of tourism destinations across different scales and types, such as coastal zones, protected areas, national parks, and even a country. These studies have generally emphasized the characteristics of natural and human systems in tourism destinations, analyzed the dynamic processes and mutual feedback mechanisms between systems, and explored the processes and mechanisms of tourism destination resilience [28,29,30]. Although there is a basic consensus to consider tourism destinations as an eco-social system, the study of destination resilience requires site-specific thinking due to the differences among tourism destinations.

Cluster #3 “COVID-19” focuses on some studies on COVID-19and tourism destination resilience. COVID-19was a global public health outbreak that had a tremendous impact on tourism destinations worldwide and generated significant academic interest in destination resilience. The representative literature of this cluster reviewed publications within the first year of the pandemic, summarized the concentrated research themes, and suggested directions for future research [31]. The impact of COVID-19on tourism and destinations has reinforced the call for resilient and sustainable tourism destinations.

Cluster #4 “system thinking” emphasizes the importance of system thinking in tourism destination resilience research. Many researchers agreed that tourism destinations should be viewed as complex systems and that tourism areas are made up of complementary products, sectors, and institutions and their interactions [32]. Social-ecological system theory and complexity theory have served as the basis for research on tourism destination resilience. Resilience and adaptability were also emphasized since destination systems are subject to change due to diverse factors, including internal system influences and external disturbances.

Cluster #5 “marine conservation” focuses on conserving marine tourism destinations. The oceans are essential but vulnerable tourism destinations. Magnus Nyström [33] found that human activities have altered the resilience of coral reefs while disturbing them and emphasized the resilience of coral reef ecosystems by focusing on the concepts of resistance, self-organization, and reorganization. Ryan Jopp [29] developed and validated a framework for adaptive management of coastal tourism sites under the background of climate change.

Cluster #6 “destination image” concerns destination image issues, mainly including the impact of disasters on destination image and ways to recover destination image. Many studies have shown that disasters and crises can harm the image of a destination, thus influencing tourists’ choice of destination [34,35]. Of course, some studies have found that developing black tourism after a disaster is also a positive direction for tourism destination development [36,37]. The representative literature of this cluster examined the changing destination image of Thailand as a tourism destination during the crisis and tested stakeholder resilience [38].

Cluster #7 “shocks” is associated with shocks to tourism destinations. The characteristics and processes of shocks and the response process of destinations must be considered in destination resilience research. Of course, different shocks have different impacts on tourism destinations, and how to measure the impact of shocks on tourism destinations is still a concern for researchers. Besides, more literature is needed to focus on the resilience of tourism destinations to specific shocks and the factors that influence this resilience. Representative literature of this cluster empirically examined coral reef perceived resilience of tourism enterprises on the Great Barrier Reef in Australia to significant disturbances or shocks. Furthermore, it demonstrated that human capital is vital for enhancing enterprise resilience [39].

Cluster #8 “research outlook” includes some literature on research outlook literature. The representative literature of this cluster focused on the disturbance and resilience of coral reef destinations. This study highlighted the shift in perspective that such studies have undergone over the last few decades, from the nature of disturbance itself to the ability of coral reefs to recover from disorder [33].

Cluster#9 “planning” relates to planning for tourism destination resilience. Destination resilience planning is an essential tool for destination management. The processes of change and their interrelationships have become more complex in a globalized and accelerating world, leading to pressure on tourism to respond and adapt to various factors. However, there is still a lack of work on tourism governance and resilience [40]. Actions should be taken at different levels of government to enable the assessment, planning, and management of long-term destination resilience [41,42].

Cluster #10 “multivariate analysis” is a cluster of literature related to resilience research methods in tourism destinations. Representative literature of this cluster employed multivariate analysis to infer the intensity of visitor impacts and predicted the resilience of tourism based on physical and biological variables [43].

Cluster #11 “total economic value” concentrates on the economic value of tourism destinations. The representative literature compared the total monetary value of ecosystem goods and services in coastal destinations in Kenya [44]. This study found that appropriate government involvement in conservation can protect high-value beach destinations. However, the absence of social and community-level values led to the loss of economic and destination resilience.

### 3.4. Co-Occurrence Analysis

According to the analysis results, this study attempts to classify the keywords into four categories. The first category refers to the study object, specifically shocks, and disasters faced by tourism destinations, such as “climate change”, “crisis”, and “risk”. The second category refers to research subjects, like “tourism”, “community”, and others. The third category refers to the research questions, including “governance”, “knowledge”, “management”, “adaptation”, “impact”, and “perspective”. The fourth category refers to research methods, like “models”, “frameworks”, and others. Observing the size of the nodes in Figure 4, it is evident that research questions and research methods were popular topics with a high level of attention.

The research themes in tourism destination resilience have changed over time (Figure 5). During 2000–2008, the study of destination resilience had not yet formed a prominent research hotspot. Resilience started to become a hot topic of interest between 2008and 2009. From 2011 to 2016, scholars generally focused on adaptability, and the close relationship between adaptability and resilience received more attention and recognition. Between 2016 and 2018, there was a gradual increase in research on destination resilience management. Recently resilience perceptions and knowledge have started to become important concerns, and of course, some other important research themes still exist. Generally speaking, the research theme of destination resilience has been constantly developed and deepened, and the concerns are more closely related to the sustainable development of tourism destinations.

## 4. Discussion

### 4.1. Hot Research Themes

With the above co-citation analysis and co-occurrence analysis, it was found that tourism destination resilience research has made certain progress over the past two decades. More scholars have joined in the study of tourism destination resilience, which to some extent indicates the important research significance of tourism destination resilience research. Tourism destination resilience research has undergone an evolutionary process from shallow to deep, gradually shifting from initial conceptual exploration to assessment methods, cognition of resilience, management, and application. The following five hot research themes were derived from a summary of the same research themes from 2000 to 2021.

#### 4.1.1. Concept and Connotation of Tourism Destination Resilience

The term “resilience” was originally a concept in the field of physics, meaning “bouncing back to its original place” [27]. Afterward, resilience was gradually introduced to the study of ecosystems, social systems, and socio-ecological systems [2,45]. Nevertheless, tourism destination resilience is a combination of resilience and tourism geography. The understanding of the concept and connotation of destination resilience has undergone an evolutionary process. Early studies considered destination resilience as the ability of tourism destinations to recover after a disturbance [43,46]. However, more studies later stressed the need to view destinations as complex systems, defining destination resilience as a destination’s ability to resist, adapt, and restructure in the face of disruptions or changes [8,47,48,49]. Compared to earlier studies interpreting the connotations of tourism destination resilience, later studies were relatively more developed and received more recognition.

Additionally, the conceptual distinction between vulnerability and resilience was a popular topic [50]. The understanding of scholars on the connotations of vulnerability and resilience can be broadly divided into two categories. One view believed that vulnerability and resilience were two sides of the same coin and were antonyms of each other. Specifically, destinations with high vulnerability had low resilience, and vice versa [48,51]. Another view held that the distinction between the two could not simply be generalized as antonyms. Both vulnerability and resilience are essential concepts in sustainable development, and destinations with high vulnerability do not necessarily lead to low resilience, which also involves other issues such as destination adaptation [5,7,52]. Vulnerability and resilience were two independent and highly synergistic concepts that were not mutually exclusive. The systematic relationship between vulnerability and resilience can help analyze tourism in protected areas in a globally changing environment [53].

#### 4.1.2. Drivers of Tourism Destination Resilience

Drivers are factors that cause changes in a system. There are many drivers of tourism destination resilience, either endogenous to the system or external perturbations that pressure the system [54,55]. In the past, the methods used to study the drivers of tourism destination resilience were mainly qualitative and quantitative, where qualitative analysis was more compared to quantitative analysis. The complexity of tourism destination systems increases the difficulty of quantitative research.

Of course, the drivers of destination resilience vary by destination. Colin Arrowsmith [43] examined the relationship between biophysical variables and the environmental resilience of the national park by adopting the principal components method, and the results showed that the resilience increased with increasing elevation. Esteban Ruiz-Ballesteros [48] found that tourism activities have a crucial influential role in the resilience of tourism communities and noted that sustainable development could only be achieved in resilient communities. Diana Kutzner [8] conducted qualitative interviews with tour operators, conservation organizations, and local governments and discovered that tour operator perceptions significantly impact the resilience of birding tourism sites. This study also proposed a conceptual framework that highlights coping strategies for operators to address perceived drivers of change. Chloe King [56] found that livelihood capital plays a vital role in tourism destination resilience and that in Indonesia, tourism planners have overly focused on the development of “high-end” tourism forms at the expense of livelihood capital, resulting in destinations that are vulnerable to damage like COVID-19.

#### 4.1.3. Sustainable Management Framework and Practices

The study of tourism destination resilience is often closely related to sustainable management. Some researchers argued that sustainability and resilience were essentially the same and that resilience is either a critical index of sustainability or a way of achieving sustainability [57]. However, other scholars hold a different view. Stephen Espiner [9] proposed a conceptual model to debate critical research on the relationship between tourism sustainability and resilience and argued that sustainability and resilience are based on different worldviews. The study concluded that sustainable tourism might mean that tourism destination systems maintain their current state over time, while resilience implies that tourism destination systems adapt to environmental complexity, uncertainty, and change. In an era of change and uncertainty, resilience may be a more appropriate framework for tourism destination management. Indeed, more scholars have incorporated resilience into sustainable management to guide the sustainable development of tourism destinations. Emma Calgaro [50] applied the destination sustainability framework to identify the integrated factors and socio-ecological processes of vulnerability and resilience within and across destinations. The practices from different tourism destinations in Thailand illustrated the differences in vulnerability and resilience because of geographical context and background. Based on an analysis of the similarities and differences between sustainability and resilience, Alan A. Lew [58] clarified the respective roles and pointed out that the new ideal tourism destination community is sustainable and resilient.

#### 4.1.4. Perception of Tourism Destination Resilience

The perception of tourism destination resilience is one of the directions of tourism destination resilience research. Perception research is an essential method for tourism destination resilience research. Scholars have conducted studies on the perceived resilience of tourism destinations in different study regions, including the Southern Alps, the Egyptian Red Sea, urban tourism sites, and others [59,60,61].

Perception studies of tourism destination resilience can be broadly divided into two categories, one to assess destination resilience and the other to understand the factors that influence destination resilience [38,54,62,63]. Patrick Joseph Holladay [63] investigated the perceptions of social and ecological resilience of residents in six tourism communities based on a scale approach, suggesting that communities need to strengthen investment and institutional capacity to control infrastructure development. Yu Ting Joanne Khew [64] assessed the contribution of infrastructure to disaster resilience through interviews with residents.

#### 4.1.5. The Resilience of the Tourism Community

The community can be seen as an element in the tourism place system and an essential type of social-ecological system. The vulnerability of communities has become more apparent in global disasters and crises. Scholars have attempted to quantify resilience in tourism communities and applied the concept of resilience to explain how to develop community-based tourism. Natural resources are significant to communities on Minnesota’s north shore of Lake Superior. Furthermore, studies have found that the effects of global climate change on the region’s natural resources have significantly impacted the livelihoods of those who rely on these resources to provide essential ecosystem services and support the regional economy [65]. Australia’s coastal tourism industry overgrew, and its excessive exploitation of cultural and natural landscapes has put pressure on the social and ecological foundations of the surrounding communities [66].

Hence, strengthening the resilience of the tourism community has become an urgent issue to be addressed. Angelo Jonas Imperiale [67] presented a social impact assessment (SIA) framework to build community resilience, divided into four stages: understanding the local context, recognizing local concerns and capacities, engaging the local community, and empowering sustainable transformation. Ladan Ghahramani [68] found that Gullah Geechee’s cultural heritage can also enhance community resilience, promote more sustainable community ownership, and suggested that the various dimensions of community loss be incorporated into decision-making. Existing studies remained dominated by qualitative analysis regarding the enhancement of resilience in tourism communities.

### 4.2. Research Gaps and Future Research Directions

Research gaps and future research directions were identified based on a combination of quantitative and qualitative analysis. After the co-citation analysis, keyword co-occurrence analysis, and research theme analysis described above, the extensive literature was analyzed and summarized. This study identified four research gaps and future research directions through qualitative analysis, including the theoretical framework of tourism destination resilience, assessment of tourism destination resilience, sustainable management and resilience, and application of advanced technology in tourism destination resilience.

#### 4.2.1. Theoretical Framework of Tourism Destination Resilience

Establishing a theoretical framework is an abstraction of the actual phenomenon and a basis for research. The theoretical framework helps distill the structural features and interrelationships of events and better understand these processes. There have been some studies in the past that have attempted to explore the theoretical framework of tourism destination resilience. Yet, these studies have generally focused on specific tourism destinations or specific disaster scenarios and lack systematic and comprehensive analyses [27,69,70]. Hence, there has not been a universally accepted theoretical framework for tourism destination resilience.

Future research on the theoretical framework of tourism destination resilience may consider the following aspects. Firstly, the tourism destination is a complex system, and it is crucial to analyze the components and characteristics of the tourism destination system. Tourism destination systems could be divided into ecological, social, economic, and cultural sub-systems. Secondly, the disturbance factors that a tourism destination system may encounter are various and complex, and the simple classification of disturbances into natural and human disturbances is still open to question. In addition, it is essential to analyze the processes in the resilience of tourism destinations. Theories including social-ecological system theory, adaptive cycle theory, chaos theory, and human-earth system theory will also provide references for studying the theoretical framework of tourism destination resilience.

#### 4.2.2. Assessment of Tourism Destination Resilience

The second research gap is the assessment of tourism destination resilience, and this work is a prerequisite for the regulation and management of tourism destination resilience. Nevertheless, the existing studies on tourism destination resilience assessment mainly were qualitative, focusing on theoretical models or assessment frameworks in practice [51]. The methods of evaluation were primarily based on qualitative methods, including semi-structured interviews and scale methods, and a few studies have attempted to quantify the degree of tourism destination resilience. Thus, these studies were generally subjective [40,55,61]. Tourism destinations are complex, typically characterized by multi-scale, multi-stage, and multi-types. Furthermore, difficulties and limitations, including data collection, will be encountered in actual assessment operations. Thus, establishing a systematic and comprehensive resilience assessment model for tourism destinations is challenging.

Several aspects of future tourism destination resilience assessment could be studied. Firstly, it is meaningful to establish an index system for tourism destination resilience assessment. Through refining the characteristics of tourism destination resilience and analyzing the systematic perturbation factors of the destination, an operable tourism destination resilience evaluation system can be developed. The ecological, social-economic, resistance, resilience, and self-organization capability aspects of tourism destinations should all be considered when choosing evaluation indicators. Secondly, it is very promising to introduce more models to assess tourism destination resilience. Models such as the Pressure-State-Response model, Driving force-State-Response model, system dynamics, structural equations, and scenario simulation may be applied to evaluate tourism destination resilience. However, it should be emphasized that the complexity of the model should not be pursued one-sidedly, but the problem-solving capability of the model should be pursued.

In addition, the assessment of tourism destination resilience should pay attention to the spatial and temporal scales. On the one hand, large- and medium-scale tourism destinations are richer in elements, and the composite indicator method can consider multidimensional factors more comprehensively. However, small-scale tourism destinations are more microscopic and experimental, and interview methods may be easier to operate. On the other hand, it is possible to better achieve destination monitoring and management by concentrating on the temporal dynamics of tourism destination resilience. It is also an issue worth exploring to conduct a prediction of tourism destination resilience.

#### 4.2.3. Sustainable Management and Resilience

The third research direction is the study of sustainable management and resilience. While several studies have analyzed the relationship between destination sustainable management and resilience, sustainable management and resilience have become complementary and synergistic [9,58]. From the definition of vulnerability and resilience, the sustainability of the destination can be achieved by reducing vulnerability and increasing resilience. However, past research has focused on conducting vulnerability assessment studies in the context of sustainable management of destinations [5,50], and not enough attention has been paid to resilience assessment.

Several areas may be worth considering in the future. Firstly, it is crucial to establish a systematic and holistic-oriented sustainable management framework that incorporates resilience assessment and management. Secondly, strengthening destination resilience in terms of destination sustainable management-oriented aspects, such as stakeholder collaboration and destination crisis management, remains a challenge. In addition, it is necessary to conduct more empirical case studies because different case sites have other conditions, and case studies can help better verify the validity of the research and be more instructive.

#### 4.2.4. Application of Advanced Technology in Tourism Destination Resilience

The fourth research direction is the application of new technologies in the resilience of tourism destinations. Especially with the advent of COVID-19, increasing attention has been paid to applying new technologies in tourism. The relationship between technology and tourism products, destination management, and visitor behavior is increasingly close. There is a gap in the application and research of technology for resilience in tourism destinations.

There are still many issues to explore with technology in the management of tourism destination resilience. At first, the application of modern technology to develop destination products like virtual tourism and these technology products can improve the diversity of tourism destination supply products. Moreover, it is meaningful to diversify product supply to enhance tourism destination resilience. Secondly, future research might analyze the role that improving the knowledge and skills of tourism practitioners in destinations plays in destination resilience and how to improve the knowledge and skills of destination staff. Thirdly, advanced technology might be utilized to monitor tourism destinations more comprehensively. On the one hand, technology can facilitate the emergency management of tourism destinations. On the other hand, it can provide multiple sources of data to assess the disaster resilience of tourism destinations.

## 5. Conclusions

Based on literature published in the WoS core collection from 2000 to 2021, the bibliometric and qualitative analyses were applied to analyze the progress of research on tourism destination resilience. A comprehensive review was achieved through collaborative network analysis, co-citation analysis, and keyword co-occurrence analysis in CiteSpace. The results revealed that the United States, Australia, China, and the United Kingdom are the leading countries for tourism destination resilience research. Both co-citation clustering networks and keyword co-occurrence networks were generated for the study. Moreover, five hot research themes were identified: (1) concept and connotation of tourism destination resilience, (2) drivers of tourism destination resilience, (3) sustainable management framework and practices, (4) perception of tourism destination resilience, (5) the resilience of the tourism community. This study also identified four research gaps and future research directions: theoretical framework of tourism destination resilience, assessment of tourism destination resilience, sustainable management and resilience, and application of advanced technology in tourism destination resilience.

Nevertheless, there are still a few aspects for improvement in this study. Only literature from the WoS database was selected for this study, which would inevitably ignore relevant literature from other databases. Future studies could try to expand the data sources to other databases to obtain a more comprehensive analysis. Nevertheless, it should also be recognized that the authority and representativeness of the WoS core collection database, to a certain extent, ensures that the results of this study are reliable. 

Overall, this study conducted a comprehensive and systematic review of tourism destination resilience research, which provided a valuable reference for future research, especially when tourism destination resilience research has become so urgent and heated.

## Figures and Tables

**Figure 1 ijerph-19-05562-f001:**
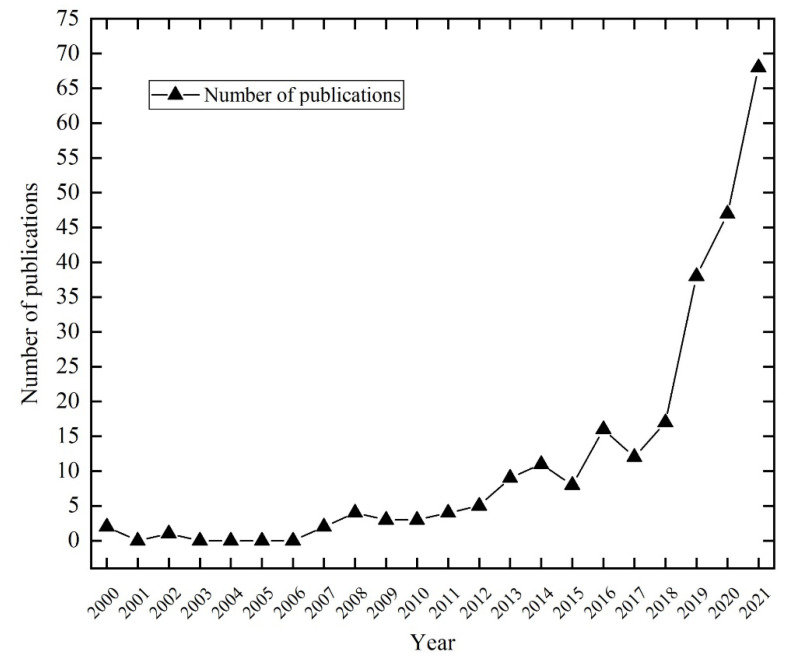
Trends in the number of publications during 2000–2021.

**Figure 2 ijerph-19-05562-f002:**
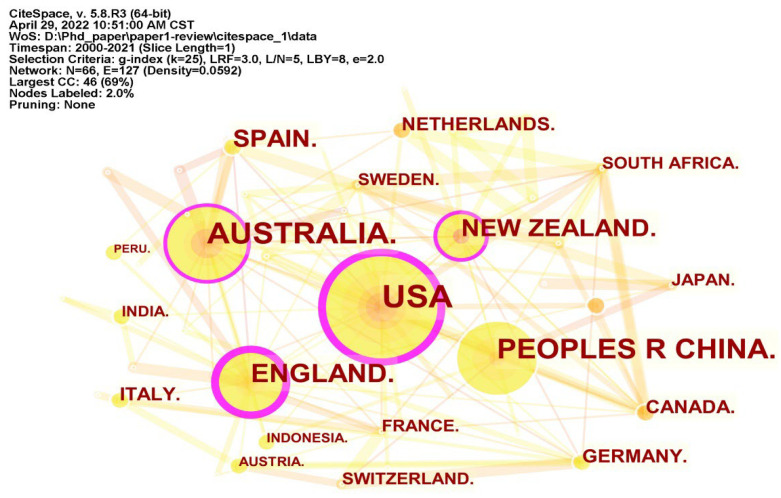
Collaborative network between countries or regions.

**Figure 3 ijerph-19-05562-f003:**
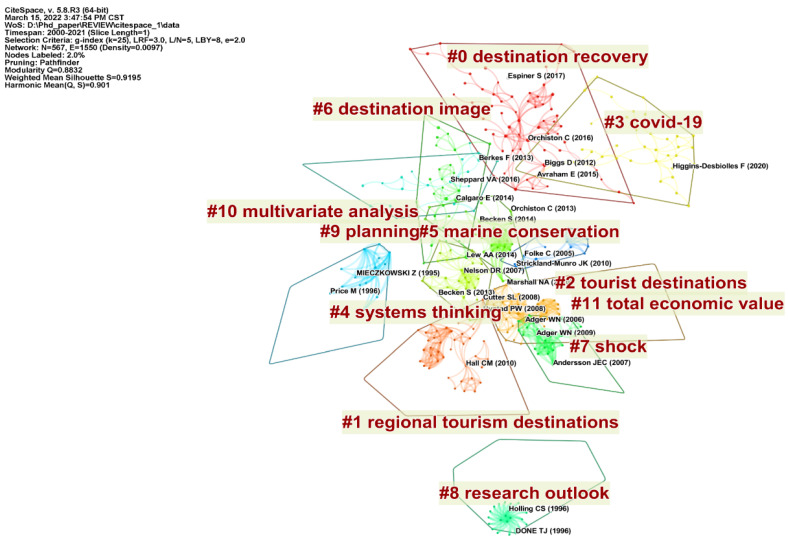
Clustering network of literature co-citations.

**Figure 4 ijerph-19-05562-f004:**
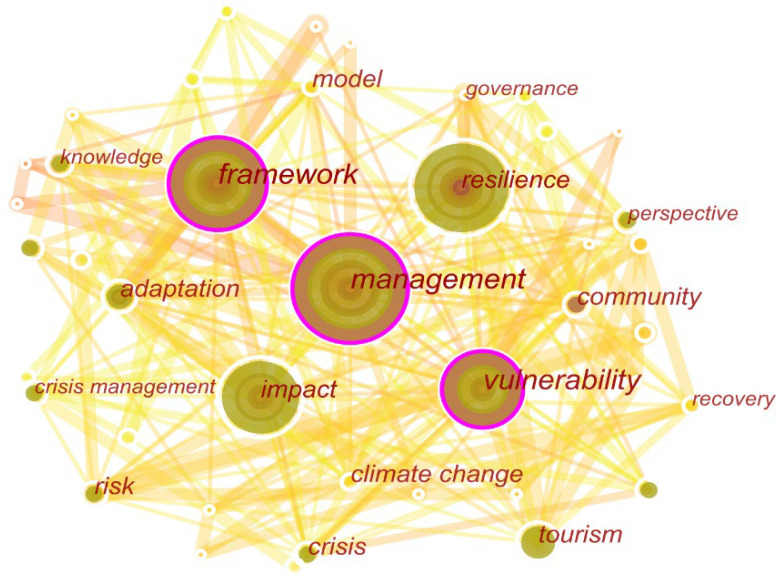
Keywords co-occurrence network during 2000–2021.

**Figure 5 ijerph-19-05562-f005:**
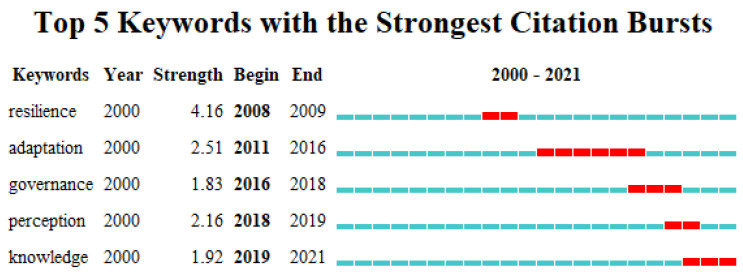
Top five keywords with the strongest citation bursts.

## Data Availability

The dataset is provided by the Web of Science dataset (https://www.webofscience.com/), accessed on 27 February 2022.

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
