# Peer review of "Bibliometric Analysis and Literature Review of Tourism Destination Resilience Research"

_ijerph, 2022, doi:10.3390/ijerph19095562_

Round 1

Reviewer 1 Report

This manuscript focuses on tourism/destination resilience literature, which is a hot topic in tourism and related fields in recent years.

This study is very well-written and organized, thus requiring only a few minor checks/revisions.

1) please write a detailed explanation about how to select/identify the target studies.

 2) it might be much more interesting if the authors can identify/describe the rise or fall of the specific topics across the time span. in the current study, a total of 11 topics were identified. how about the trend of each of 11 topics?

Reviewer 2 Report

The topic of destination resilience is indeed among issues that are more and more commonly studied by scientists representing different areas of knowledge. The growing body of literature allows at the moment conducting of the first literature studies and bibliometric analysis. Definitely, the selection of the topic is justified properly. The bibliometric analysis is prepared well. Due to the high number of analyzed sources adding the table covering all sources is not necessary and desired. As usual, the selection of the database for such an analysis can be controversial, but the Authors know about this and present clear arguments as well as the limitations of their choice. The value of the paper is amended by attached figures presenting the results of their network and clustering analysis. If I should find some weaknesses of the paper, I would search for them in the conclusions. This part is rather short, but it contains the most expected statements in it.

Reviewer 3 Report

I believe that the topic is absolutely timely, not only, but especially considering the COVID-19 pandemic. The database used (WoS) is no doubt adequate. All research seems consistent, following a well-explained methodology. Results are shared, as well as some future hypotheses to continue this study. I am absolutely comfortable recommending this paper for publication.

Reviewer 4 Report

The article is interesting but I have a few suggestions for a better comprehension:

  • lines 89-90: review the phrase
  • A better explanation of the time span should be useful. 
  • The quality of figure 2 is poor. 
  • The discussion section includes new ideas whose origin is not clear, i.e. DPSIR.
